# APPROXIMATELY PIECEWISE E(3) EQUIVARIANT POINT NETWORKS

**Matan Atzmon** [1] **Jiahui Huang** [1] **Francis Williams** [1] **Or Litany** [1] [2]

[1] NVIDIA   [2] Technion

`{matzmon,jiahuih,fwilliams,olitany}@nvidia.com`

## ABSTRACT

Integrating a notion of symmetry into point cloud neural networks is a provably effective way to improve their generalization capability. Of particular interest are $E(3)$ equivariant point cloud networks where Euclidean transformations applied to the inputs are preserved in the outputs. Recent efforts aim to extend networks that are equivariant with respect to a single global $E(3)$ transformation, to accommodate inputs made of multiple parts, each of which exhibits local $E(3)$ symmetry. In practical settings, however, the partitioning into individually transforming regions is *unknown* a priori. Errors in the partition prediction would unavoidably map to errors in respecting the true input symmetry. Past works have proposed different ways to predict the partition, which may exhibit uncontrolled errors in their ability to maintain equivariance to the actual partition. To this end, we introduce APEN: a general framework for constructing approximate piecewise-$E(3)$ equivariant point networks. Our framework offers an adaptable design to *guaranteed* bounds on the resulting piecewise $E(3)$ equivariance approximation errors. Our primary insight is that functions which are equivariant with respect to a *finer* partition (compared to the unknown true partition) will also maintain equivariance in relation to the true partition. Leveraging this observation, we propose a compositional design for a partition prediction model. It initiates with a fine partition and incrementally transitions towards a coarser subpartition of the true one, consistently maintaining piecewise equivariance in relation to the current partition. As a result, the equivariance approximation error can be bounded solely in terms of (i) uncertainty quantification of the partition prediction, and (ii) bounds on the probability of failing to suggest a proper subpartition of the ground truth one. We demonstrate the practical effectiveness of APEN using two data types exemplifying part-based symmetry: (i) real-world scans of room scenes containing multiple furniture-type objects; and, (ii) human motions, characterized by articulated parts exhibiting rigid movement. Our empirical results demonstrate the advantage of integrating piecewise $E(3)$ symmetry into network design, showing a distinct improvement in generalization accuracy compared to prior works for both classification and segmentation tasks.

## 1 INTRODUCTION

In recent years, there has been an ongoing research effort on the modeling of neural networks for 3D recognition tasks. Point clouds, as a simple and prevalent 3D input representation, have received substantial focus, leading to *point networks*: specialized neural network architectures operating on point clouds (Qi et al., 2017; Zaheer et al., 2017). Since many point cloud recognition tasks can be characterized as equivariant functions, modeling them with an equivariant point network has been shown to be an effective approach. Indeed, equivariant modeling can simplify a learning problem: knowledge learned from one input, automatically propagates to all input's symmetries(Bietti et al., 2021; Elesedy & Zaidi, 2021; Tahmasebi & Jegelka, 2023).

One important symmetry exhibited in point clouds is the *Euclidean motions*, $E(3)$, consisting of all the possible rigid motions in space. Building on the demonstrated success of $E(3)$ equivariant point networks in prior research (Thomas et al., 2018), recent efforts have been dedicated to

extending $E(3)$ symmetry to model *piecewise* rigid motions symmetry as well (Yu et al., 2022; Lei et al., 2023; Deng et al., 2023). This extension is valuable since some recognition tasks can be better characterized as piecewise $E(3)$ equivariant functions. To support this claim, we turn to the task of instance segmentation within a scene, illustrated by a 2D toy example in the right inset.

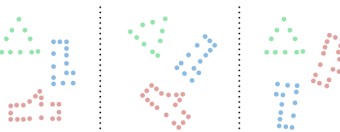

In the leftmost column, we visualize segmentation predictions by distinct colors. In the middle column, we observe the expected invariant predictions under a global Euclidean motion of the entire scene. Finally, in the right column, we showcase invariant predictions under a *piecewise* deformation that allows individual objects to move independently in a rigid manner, decoupled from the overall scene's motion.

Incorporating piecewise $E(3)$ symmetry to point networks presents several challenges. The primary hurdle is the unknown partitioning of the input point cloud into its moving parts. While having such a partition makes it possible to implement equivariant design using a $E(3)$ equivariant *siamese* network across parts (Atzmon et al., 2022), this is often infeasible in real-world applications. For instance, in the segmentation task shown in the inset, the partition is inherently tied to the model's segmentation predictions. Thus, in cases where the underlying partition is not predefined but rather predicted by a (non-degenerated) model, any suggested piecewise equivariant model will introduce an *approximation error* in satisfying the equivariance constraint. We will use the term *equivariance approximation error* to refer to the error that arises when a function is unable to satisfy the piecewise $E(3)$ equivariance constraint (w.r.t. the true unknown partition); see Definition 1.This equivariance approximation error is inherent unless the partition prediction remains perfectly consistent under the input symmetries. This implies it must be invariant to the very partition it seeks to identify. So far in the literature, less attention has been given to piecewise equivariant network designs that offer means to control the network's equivariance approximation error. For example, Liu et al. (2023) suggests an initial partition prediction model based on input points' global $E(3)$ invariant and equivariant features. In Yu et al. (2022), local-context invariant features are used for the partition prediction model. In both cases, it is unclear how failures in the underlying partition prediction model will affect the equivariance approximation error. Notably, the concurrent work of Deng et al. (2023) also observes the equivariance approximation error. Their work suggests an *optimization-based* partition prediction model based on (approximately) contractive steps, striving to achieve exact piecewise equivariance; errors in the partition prediction model arising from expanding steps and their impact on the resulting equivariance approximation error are not discussed.

In this paper, we propose a novel framework for the design of approximately piecewise equivariant networks, called APEN. Our goal is to suggest a practical design that can serve as a backbone for piecewise $E(3)$ equivariant tasks, while identifying how elements in the design control the piecewise equivariance approximation error. Our framework is built on the following simple fact. Let $G$ and $G'$ be two symmetry groups for which each symmetry in $G'$ is also in $G$, i.e., $G' \leqslant G$. Then, any $G$ equivariant function is also a $G'$ equivariant function. Thus, we can have an *exact* piecewise equivariant model, as long as the model partition is a proper subpartition of the (unknown) ground-truth one.

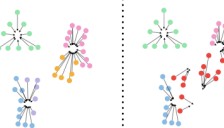

The right inset illustrates this fact: the piecewise equivariant predictions of vote targets, marked as black dots, are accurate for a subpartition of the ground-truth partition (left column), whereas an equivariant approximation error arises for a partition that includes a *bad* part consisting of points mixed from two different parts in the ground truth partition (i.e., the red dots in the right column). This observation may lead to the following *simple* model for partition prediction – drawing a random partition from the distribution of non-degenerated partitions of size $k$ (i.e., all $k$ parts get at least one point). For such a model, the probability of drawing a bad part reaches 0 as $k$ increases. In turn, the probability of drawing a bad partition can be used to bound the equivariance approximation error of a piecewise equivariant function, as good sub-partitions induce *no* equivariance approximation error. Importantly, this approach alleviates the need for additional constraints on the underlying model function to control the equivariance approximation error.

However, this approach needs to be pursued with caution, as increasing the complexity of the possible partitions reduces the expressivity of the resulting piecewise equivariant point network model class. This caveat is especially relevant to the common design using a *shared* (among parts) $E(3)$ equivariant backbone. Indeed, at the limit where each point belongs to a distinct part, the only shared backbone $E(3)$ equivariant functions are constant. To mitigate potential expressivity issues, our APEN framework employs a compositional network architecture. This architecture comprises a

sequence of piecewise equivariant layers, with the complexity of their underlying partition decreasing gradually. Each layer is defined as a piecewise $E(3)$ equivariant function, which not only predicts layer-specific features but also parametrizes a prediction of a *coarser* partition. This coarser partition serves as the basis for the subsequent layer's piecewise $E(3)$ symmetry group. The goal of this "bottom-up" approach is to allow the network to overcome the issue of ambiguous predictions in earlier layers by learning to merge parts that are likely to transform together, resulting in a simpler partition in the subsequent equivariant layer. Importantly, this design also provides bounds for the piecewise equivariant approximation error of each layer, resulting solely from two sources in the design: (i) uncertainty in the partition prediction model, and (ii) the probability of drawing a bad partition.

We instantiated our APEN framework for two different recognition tasks: classification and part segmentation. We conducted experiments using datasets comprising of (i) articulated objects consisting of human subjects performing various sequence movements (Bogo et al., 2017), and (ii) real-world room scans of furniture-type objects (Huang et al., 2021a). The results validate the efficacy of our framework and support the notion of potential benefits in incorporating piecewise $E(3)$ deformations to point networks.

## 2 METHOD

### 2.1 BACKGROUND: EQUIVARIANT POINT NETWORKS

We will consider point networks as functions $h : U \to W$, where $U$ and $W$ denote the vector spaces for the input and output domains, respectively. The input vector space $U$ takes the form $U = \mathbb{R}^{n \times (2 \times d)}$, with $n$ denoting the number of points in the input point cloud, $d$ is the point embedding space dimension (usually $d = 3$), and 2 per-point features: spatial location and an *oriented normal* vector. Depending on the task at hand, classification, or segmentation, the output vector space $W$ can be $W = \mathbb{R}^c$ or $W = \mathbb{R}^{n \times c}$. To incorporate symmetries into a point network, we consider a group $G$, along with its action $g$ on the vector spaces $U$ and $W$. Of particular interest in our work is the Euclidean motions group $G = E(d)$ defined by rotations, reflections and translations in $d$-dimensional space. The group action on $\boldsymbol{X} \in U$ is defined by $g \cdot \boldsymbol{X} = \boldsymbol{X}\boldsymbol{R}^T + \mathbf{1}\boldsymbol{t}^T$, with $g = (\boldsymbol{R}, \boldsymbol{t})$ being an element in $E(d)$[1], while the action on the output $\boldsymbol{Y} \in W$ varies depending on the task (e.g., $g \cdot \boldsymbol{Y} = \boldsymbol{Y}$ for classification). An important property for our networks $h$ to satisfy is *equivariance* with respect to $G$:

$$h(g \cdot \boldsymbol{X}) = g \cdot h(\boldsymbol{X}) \qquad \forall g \in G, \boldsymbol{X} \in U. \tag{1}$$

We consider the typical case of networks $h$ which follow an *encoder-decoder* structure, i.e., $h = \mathtt{d} \circ \mathtt{e}$. The encoder $\mathtt{e} : U \to V$ transforms an input into a learnable latent representation $V$. In our case, $V$ is an $E(3)$ equivariant latent space, up to order type 1, of the form $V = \mathbb{R}^{a+b \times 3}$, with $a, b$ being positive integers. The decoder $\mathtt{d} : V \to W$ decodes the latent representation to produce the expected output response which can be invariant or equivariant to the input. Both $\mathtt{e}$ and $\mathtt{d}$ are modeled as a composition of multiple invariant or equivariant layers. Having covered the basics of equivariant point networks, we will now proceed to describe our proposed framework, starting with the formulation of a piecewise $E(d)$ equivariant layer.

### 2.2 PIECEWISE $E(d)$ EQUIVARIANCE LAYER

We start this section by describing the settings for which we model a piecewise $E(d)$ equivariant layer. Let $\boldsymbol{X} \in U$ be the input to the layer. Our assumption is that the partition prediction is modeled as a (conditional) probability distribution, $Q_{\boldsymbol{Z}|\boldsymbol{X}} \in (\Sigma_k)^n$ over the $k$ parts partitions $\boldsymbol{X}$ can exhibit. Here $\Sigma_k$ denotes the $k$ probability simplex. Let $\boldsymbol{Z} = \left[\boldsymbol{z}_1^T, \cdots, \boldsymbol{z}_n^T\right]^T \in \{0,1\}^{n \times k}$ with $\boldsymbol{Z}\mathbf{1} = \mathbf{1}$, denote a realization of a partition from $Q_{\boldsymbol{Z}|\boldsymbol{X}}$, i.e., $\boldsymbol{Z} \sim Q_{\boldsymbol{Z}|\boldsymbol{X}}$.

Let $\widehat{\boldsymbol{Z}}$ be the *unknown* ground truth partition of $\boldsymbol{X}$. An important quantity of interest is

$$\lambda(Q) = P_{\boldsymbol{Z} \sim Q_{\boldsymbol{Z}|\boldsymbol{X}}} \left( \exists\, 1 \le i, j \le n \text{ s.t. } (\boldsymbol{Z}\boldsymbol{Z}^T)_{ij} > (\widehat{\boldsymbol{Z}}\widehat{\boldsymbol{Z}}^T)_{ij} \right), \tag{2}$$

---

[1]Note that in fact $g = (\boldsymbol{R}, \mathbf{0})$ on the input normals features.

measuring the probability of drawing a "bad" partition from $Q$, i.e., a non-proper subpartition of $\widehat{Z}$. In that context, a reference partition prediction model is $Q_{\text{simple}}$ which is defined by a uniform draw of a partition satisfying $\mathbf{1}^T Z e_j > 0$ for each $j \in [k]$. An important property of $Q_{\text{simple}}$ is $\lambda(Q_{\text{simple}}) \to 0$ as $k \to n$. To better understand this claim about $\lambda(Q_{\text{simple}})$, one can consider the sequential process generating a random $k$ parts partition. Clearly, larger values of $k$ result in each part containing fewer points. Since the probability of drawing the next point from mixed ground-truth parts is independent of $k$, determined solely by the number of input points and the ground-truth partition, the probability that the next drawn point generated a bad part lowers as $k$ increases. In turn, $\lambda(Q_{\text{simple}})$ can serve as a useful bound for the resulting equivariance approximation error. Consequently, we opt for a model $Q$ that satisfies $\limsup \lambda(Q) = \lambda(Q_{\text{simple}})$, where the last limit is taken with respect to a hyper-parameter in the design of $Q$.

More precisely, we suggest the following characterization for $Q$. Let $\delta : (\Sigma_k)^n \to \mathbb{R}_+$, satisfying

$$\delta(Q) \to 0, \text{ whenever } Q \to Q_v \tag{3}$$

with $Q_v \in \{0, 1\}^{n \times k} \cap (\Sigma_k)^n$. That is, $\delta$ measures the uncertainty in the model's prediction. Our design *requirement* is that

$$\limsup \lambda(Q) = \lambda(Q_{\text{simple}}), \text{ as } \delta(Q) \to 0. \tag{4}$$

In other words, we suggest constraining a $Q$ model to behave in a way such that, as it becomes more *certain* in how it draws its predictions, the probability of drawing a "bad" partition converges to be *no worse* than the one of the simple model. The functional $\delta$ measures the uncertainty of $Q$, and is considered as one of the design choices in the modeling of $Q$. In turn, it will be used to bound the equivariance approximation error. Fig. 1 illustrates the qualitative behavior of $\delta$.

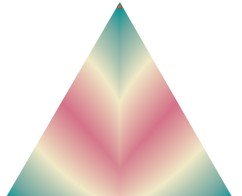

Figure 1: The functional bound $\delta$. Green colors indicate values close to $0$.

We defer the discussion on how we provide a model for $Q$ supporting these ideas for later. Instead, we start by describing how $Q_{Z|X}$ is incorporated to model a piecewise equivariant layer.

**Fixed partition.** To facilitate discussion, we first assume that $Z$ is fixed, and we will start by describing a piecewise $E(d)$ equivariant layer with respect to $Z$ partition. Let $G = E(d) \times \cdots \times E(d)$ be the product consisting of $k$ copies of the Euclidean motions group. For $g = (g_1 \cdots, g_k) \in G$, we define

$$g \cdot (X, Z) = \sum_{j=1}^{k} (g_j \cdot X) \odot (Z e_j \mathbf{1}_d^T), \tag{5}$$

where $g_j \cdot X = X R_j^T + \mathbf{1}_n t_j^T$, $\{e_j\}_{j=1}^k$ is the standard basis in $\mathbb{R}^k$, $\mathbf{1}_d$ is the vector of all ones in $\mathbb{R}^d$, and $\odot$ denotes the Hadamard product between two matrices.

One appealing way to model a piecewise $E(d)$ equivariant function, $\psi : U \times \{0, 1\}^{n \times k} \to U'$, which also respects the inherited order symmetry of the part's assignments, is by employing an $E(d)$-equivariant backbone $\psi_b : U \to U'$ shared among the parts (Atzmon et al., 2022; Deng et al., 2023), taking the form:

$$\psi(X, Z) = \sum_{j=1}^{k} \psi_b(X \odot Z e_j \mathbf{1}_d^T) \odot Z e_j \mathbf{1}^T. \tag{6}$$

The following lemma, whose proof can be found in the Appendix, verifies these properties for $\psi$.

**Lemma 1.** *Let $\psi : U \times \{0, 1\}^{n \times k} \to U'$ be a function as in Eq. (6). Let $g \in G$ and $\sigma_k(\cdot)$ a permutation on $[k]$. Then,*

$$\psi(g \cdot (X, Z), Z) = g \cdot (\psi(X, Z), Z),$$
$$\psi(X, Z') = \psi(X, Z)$$

*for any $X \in U$, $Z \in \{0, 1\}^{n \times k}$, and $Z' = Z_{:, \sigma(i)}$.*

Note that one can consider augmenting the design of Eq. (6) with a function over orderless representation of parts $E(d)$ invariant features (Maron et al., 2020). Equipped with the construction in Eq. (6), we will now move on to the case where $Z$ is uncertain.

**Uncertain partition.** Incorporating $Q_{\boldsymbol{Z}|\boldsymbol{X}}$ into a layer can be done by marginalizing over the possible $\boldsymbol{Z}$. Some simple options for marginalization are i) $\phi_{\mathrm{I}}(\boldsymbol{X}) = \psi(\boldsymbol{X}, \mathbb{E}_Q \boldsymbol{Z})$ as implemented in Atzmon et al. (2022); ii) $\phi_{\mathrm{II}}(\boldsymbol{X}) = \mathbb{E}_Q \psi(\boldsymbol{X}, \boldsymbol{Z})$; and iii) $\phi_{\mathrm{III}}(\boldsymbol{X}) = \psi(\boldsymbol{X}, \boldsymbol{Z}_*)$, where $(\boldsymbol{Z}_*)_{i,:} = \boldsymbol{e}_{\arg\max_j Q(\boldsymbol{Z}|\boldsymbol{X})_{ij}}$. Unfortunately, however, all of these options are merely an *approximation* of a piecewise $E(d)$ equivariant function. The scheme $\phi_{\mathrm{I}}$ relies on scaling, which can be an arbitrarily bad approximation to the input's geometry. The scheme $\phi_{\mathrm{II}}$ relies on the averaging of equivariant point *features*, which is not stable under a realization of a particular partition $\boldsymbol{Z} \sim Q$. Similarly, $\phi_{\mathrm{III}}$ is also not equivariant under all possible realizations of $\boldsymbol{Z}$. However, the equivariance approximation error $\phi_{\mathrm{III}}$ induces can be controlled, as we discuss next.

**Bounding the equivariant approximation error.** In this work, we advocate for layers of the form $\phi_{\mathrm{III}}$. The motivation for doing so is that it enables a *uniform* control over the equivariant approximation error as a function of $Q$, crucially, without relying on bounding the *variation* of $\phi$. This advantage is especially prominent for neural networks, as existing techniques for bounding network's bounded variation, e.g., by controlling the network's Lipshitz constant, impose additional complexity to the network architecture and may hinder the training process (Anil et al., 2019). On the other hand, as we will see in the next section, the approximation error $Q$ induces can be controlled explicitly by a choice of hyper-parameters in the parametrization of $Q$.

The next definition captures our suggested characterization for an approximation error of a desired piecewise $E(d)$ equivariant layer:

**Definition 1.** *Let $\phi : U \to U'$ be a bounded function with $\|\phi\| \le M$. Let $\delta : (\Sigma_k)^n \to \mathbb{R}_+$, satisfying Eq. (3) and Eq. (4) w.r.t. $Q$. Then, $\phi$ is a $(G, Q)$ equivariant function if and only if for any given $\boldsymbol{X} \in U$, the following is satisfied*

$$\mathbb{E}_{Q_{\boldsymbol{Z}|\boldsymbol{X}}} \|\phi(g \cdot (\boldsymbol{X}, \boldsymbol{Z})) - g \cdot (\phi(\boldsymbol{X}), \boldsymbol{Z})\| \le (\lambda(Q_{simple}) + \delta(Q)) M \tag{7}$$

*for all $g \in G$. We denote the set of $(G, Q)$ equivariant functions by $\mathcal{F}_Q$.*

The above characterization for the equivariance approximation error can be seen as resulting from two different sources of properties in the partition prediction model: (i) an intrinsic source, as captured by $\delta$, which measures the uncertainty of the model $Q$, and (ii) an extrinsic source, determined by a measure independent from $Q$ as captured by $\lambda$. In addition, the above definition generalizes the notion of exact equivariant function classes. For instance, consider $\widehat{\boldsymbol{Z}}$ satisfying $\widehat{\boldsymbol{Z}} \boldsymbol{e}_j = \mathbf{1}$ for some *fixed* $j$; setting $\delta \equiv 0$ yields that $\mathcal{F}_Q$ coincides with the class of global $E(d)$ equivariant functions.

To conclude this section, we verify in the following theorem that our construction of $\phi$ indeed falls under the suggested characterization of approximate piecewise $E(d)$ equivariant functions. Proof details are in the Appendix.

**Theorem 1.** *Let $\phi : U \to U'$ be of the form*

$$\phi(\boldsymbol{X}) = \sum_{j=1}^k \psi_b(\boldsymbol{X} \odot \boldsymbol{Z}_* \boldsymbol{e}_j \mathbf{1}_d^T) \odot \boldsymbol{Z}_* \boldsymbol{e}_j \mathbf{1}^T, \tag{8}$$

*where $(\boldsymbol{Z}_*)_{i,:} = \boldsymbol{e}_{\arg\max_j Q(\boldsymbol{Z}|\boldsymbol{X})_{ij}}$, and $\psi_b : U \to U'$ is an $E(d)$ equivariant backbone. Then,*

$$\phi \in \mathcal{F}_Q.$$

## 2.3 Q PREDICTION

So far, we have treated $Q$ as a given input to the layer. In fact, we suggest that $Q$ results from a piecewise equivariant prediction of a prior layer. Exceptional is the first layer, for which $Q = Q_{\mathrm{simple}}$. Given a layer output of the form in Eq. (8), we will next describe how $Q^{\mathrm{pred}}$ is inferred. Note that $Q$ still denotes the given input partition prediction model.

**Modeling considerations.** As a first attempt, one might consider parametrizing $Q^{\mathrm{pred}}$ as the $\mathrm{softmax}$ of a per-point $Q$ piecewise invariant layer prediction. However, this approach introduces several difficulties, causing it to be unfeasible. Firstly, it is unclear how to supervise $Q$ during training to predict good sub-partitions of the ground-truth partition. Secondly, network optimization could be tricky, since the domain of possible partition solutions has a high dimensional combinatorial structure,

especially due to our design bias for a large number of parts in early network layers. Lastly, there is a need to model the merging of parts in the input partition to generate a coarser one.

To address these challenges, we propose a geometric approach to model $Q^{\text{pred}}$. Our suggestion is to set $Q^{\text{pred}}$ as the assignment scores resulting from the partitioning (i.e., clustering) in $R^d$ of $Q$ piecewise *equivariant* per-point predictions. Notably, this suggestion falls under the well-known attention layer (Vaswani et al., 2017; Locatello et al., 2020; Liu et al., 2023) following a query, key, and value structure with $\phi(\boldsymbol{X})$ being the values and queries, part centers being the keys, and the prediction $Q^{\text{pred}}$ is proportional to the matching score of a query to a key. One of the advantages of this approach is that $Q^{\text{pred}}$ emerges as an *orderless* prediction with respect to possible parts assignments, thus simplifying the optimization domain. However, it is not clear how this model can (i) control the resulting $\delta(Q^{\text{pred}})$ by means of its design; and (ii) support the merging of parts to constitute a prediction of a coarser partition. To this end, we suggest that the part center (keys) predictions are set as the minimizers of an energy that is invariant to $Q$ piecewise $E(d)$ deformations of $\phi(\boldsymbol{X})$ (values). We formalize this idea in the next paragraph.

**Q Prediction.** Let $\boldsymbol{Y} = [\boldsymbol{y}_1, \cdots, \boldsymbol{y}_n]^T \in \mathbb{R}^{n \times d}$ denote the first equivariant per-point prediction in $\phi(\boldsymbol{X}) \in U'$. Let $\left[\boldsymbol{\mu}_j^*\right]_{j=1}^k \in \mathbb{R}^{d \times k}$ denote the underlying predicted part centers with which the score of $Q^{\text{pred}}$ is defined. We define $\left[\boldsymbol{\mu}_j^*\right]_{j=1}^k$ as the minimizers of an energy consisting of the negative log-likelihood of a Gaussian Mixture Model and a regularization term that constraints the KL distance between all pairs of Gaussians to be greater than some threshold. Let $P(\boldsymbol{Y}; \boldsymbol{\alpha} = (\boldsymbol{\mu}_j, \pi_j; \sigma)_{j=1}^k)$ denote the mixture distribution, parametrized by $\boldsymbol{\alpha}$. Then, the log-likelihood is $\log P(\boldsymbol{Y}; \boldsymbol{\alpha}) = \sum_{i=1}^n \log(\sum_{j=1}^k \pi_j \mathcal{N}(\boldsymbol{y}_i; \boldsymbol{\mu}_j, \sigma))$ where $\mathcal{N}(\cdot; \boldsymbol{\mu}_j, \sigma)$ denotes the density of an isotropic Gaussian random variable, centered at $\boldsymbol{\mu}_j$ with variance $\sigma^2 I$. Note that $\sigma$ is fixed and is considered as a hyper-parameter. Then, $\left[\boldsymbol{\mu}_j^*\right]_{j=1}^k$ are defined as

$$(\boldsymbol{\mu}_j^*, \pi_j^*) = \arg\min_{\boldsymbol{\alpha}} -\log P(\boldsymbol{Y}; \boldsymbol{\alpha}) - \tau \sum_{j \neq j'} \pi_j \pi_j' \log D_{\text{KL}}(\mathcal{N}(\cdot; \boldsymbol{\mu}_j) \| \mathcal{N}(\cdot; \boldsymbol{\mu}_j')). \quad (9)$$

In turn, the prediction $Q_{ij}^{\text{pred}}$ is defined as

$$Q_{ij}^{\text{pred}} = \frac{\mathcal{N}(\boldsymbol{y}_i; \boldsymbol{\mu}_j^*, \sigma)\pi_j^*}{\sum_{j=1}^k \mathcal{N}(\boldsymbol{y}_i; \boldsymbol{\mu}_j^*, \sigma)\pi_j^*}. \quad (10)$$

Importantly, the above construction yields that as $\sigma \to 0$: i) $\lambda(Q^{\text{pred}}) \to \lambda(Q_{\text{simple}})$ since each random partition is a minimizer of the likelihood functional, and ii) $\delta(Q^{\text{pred}}) \to 0$. In addition, $\sigma$ also controls the sensitivity of Gaussians to merge (under a fixed coefficient $\tau$), where larger values encourage Gaussians to explain wider distribution of values $\boldsymbol{y}_i$. Thus, setting an increasing sequence of $\sigma$ values across layers supports the gradual coarsening of partitions design. Lastly, note that differentiating the prediction of $Q^{\text{pred}}$ w.r.t. its inputs is not trivial; these details are covered in the next section.

### 2.4 IMPLEMENTATION DETAILS

**Network architecture.** We start by sharing the details about the construction of the layer $\psi$ in Eq. (6) given a known partition $\boldsymbol{Z}$. For that end, we used Frame Averaging (FA) (Puny et al., 2022) with a shared pointnet (Qi et al., 2017) network, $\tilde{\psi}$. We define our shared equivariant backbone by

$$\psi_b(\boldsymbol{X} \odot \boldsymbol{Z}\boldsymbol{e}_j \boldsymbol{1}_d^T) = \left\langle \tilde{\psi}(\boldsymbol{X} \odot \boldsymbol{Z}\boldsymbol{e}_j \boldsymbol{1}_d^T) \right\rangle_{F(\boldsymbol{X} \odot \boldsymbol{Z}\boldsymbol{e}_j \boldsymbol{1}_d^T)}$$

where $F(\boldsymbol{X} \odot \boldsymbol{Z}\boldsymbol{e}_j \boldsymbol{1}_d^T)$ is the same PCA based construction for an $E(d)$ frame suggested in Puny et al. (2022), and $\langle \cdot \rangle$ is the FA symmetrization operator. Then, $\psi(\boldsymbol{X}, \boldsymbol{Z})$ is defined exactly as in Eq. (6). Since this construction needs to support layers with a relatively large number of parts $k$, we implement the network $\psi_b$ using the sparse linear layers from Choy et al. (2019). In all our experiments, we implemented the encoder as a composition of $L$ layers, $e = \phi_L \circ \cdots \circ \phi_1$, with $L = 4$; see Fig. 2. $Q_{\text{simple}}$ is set as the input to $\phi_1$. In fact, $Q_{\text{simple}}$ can be further regulated than the naive suggestion. In practice, we set $Q_{\text{simple}}$ by a Voronoi partition resulting from $k$ furthest point samples from the input $\boldsymbol{X}$. The exact analysis of $\lambda(Q_{\text{simple}})$ as a function of $n$ and $k$ is out of the scope of this work – we only rely on Eq. (4).

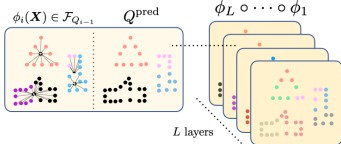

Figure 2: APEN network design.

**Q prediction.** For finding a minimizer of Eq. (9), we used a slight modification of the well-known EM algorithm (Dempster et al., 1977) that supports the merging of centers closer than the threshold $\tau$. Note that during training, the *backward* calculation requires the derivative of $\frac{\partial Q}{\partial \phi}$. Since the EM is in an iterative algorithm, this might unnecessarily increase the computational graph of the backward computation. To mitigate this, we use the following construction, based on implicit differentiation (Atzmon et al., 2019; Bai et al., 2019). Let $\tilde{\alpha}$ be a minimizer Eq. (9) that is detached from the computational graph and $Y$. Then, $s(Y; \tilde{\alpha}) = 0$ where $s(Y; \tilde{\alpha}) = \nabla_{\alpha} \log P(Y; \alpha)$, known in the literature as the score function (Bishop & Nasrabadi, 2006). We define

$$\alpha = \tilde{\alpha} + I^{-1}(\tilde{\alpha}) s(Y; \tilde{\alpha}), \tag{11}$$

where $I^{-1}(\tilde{\alpha}) = \text{Var}(s(Y; \tilde{\alpha}))$ is the fisher information matrix (Bishop & Nasrabadi, 2006) calculated at $\tilde{\alpha}$. Importantly, $I$ only depends on $s$ and does not involve second derivative calculations. It can be easily verified that $\alpha$ is a minimizer of Eq. (9) and that $\frac{\partial \alpha}{\partial Y} = \frac{\partial(\arg\min_{\alpha}(E(\alpha, Y)))}{\partial Y}$, where $E(\cdot)$ denotes the energy defined in Eq. (9). This is summarized in Alg. 1, found in the Appendix.

**Training details.** Our framework requires supervision in order to train $Q^{\text{pred}}$ to approximate the ground-truth partition. To that end, we compute the ground-truth $Y_{\text{GT}} \in \mathbb{R}^{n \times d}$ to supervise the parts center vote predictions $Y_l \in \mathbb{R}^{n \times d}$ of the $l^{\text{th}}$ layer. We utilize the given segmentation information, to calculate $Y_{\text{GT}} = ZC^T - X$, where $Z \in \{0, 1\}^{n \times k}$ are the ground-truth assignments of $X \in \mathbb{R}^{n \times d}$ and $C \in \mathbb{R}^{d \times k}$ is calculated as the center of the minimal bounding box encompassing each of the input parts. Then, a standard $L_1$ loss is added to optimization,

$$\text{loss}_{\text{A}} = \sum_{l=1}^{L} \|Y_l - Y_{\text{GT}}\|.$$

## 3 EXPERIMENTS

We evaluate our method on two types of datasets that fit piecewise $E(3)$ symmetry: (i) scans of human subjects performing various sequences of movements (Loper et al., 2015; Bogo et al., 2017; Mahmood et al., 2019), and (ii) real-world rooms scans of furniture-type objects (Huang et al., 2021a). In all of our experiments, we used the ground-truth segmentation maps to extract $Y_{\text{GT}}$ supervision as described in Sec. 2.4.

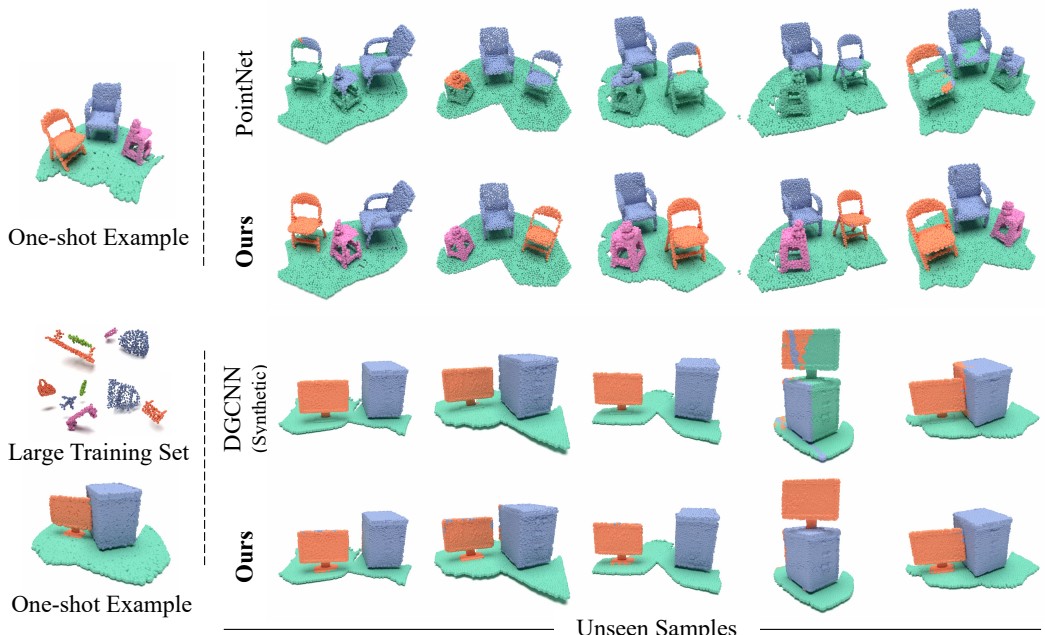

Figure 3: Qualitative results for one-shot generalization on DynLab dataset (Huang et al., 2021a).

## 3.1 HUMAN SCANS

We start by evaluating our framework for the task of point part segmentation, a basic computer-vision task with many downstream applications. Specifically, we consider human body parts segmentation, where the goal is to assign each of the input scan points to a part chosen from a predefined list. In our case, the list consists of 24 body parts.

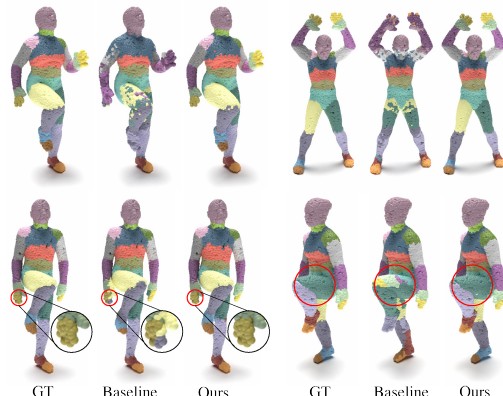

To evaluate different aspects of our framework, we use three different train/test splits. The first consists of a random (90%/10%) train/test split of 41, 461 human models from the SMPL dataset (Loper et al., 2015) consisting of 10 different human subjects as in (Huang et al., 2021b). This experiment acts as a sanity test and ensure our method does not underperform compared to baselines. The second and third splits use the scans from the Dynamic FAUST (DFAUST) dataset (Bogo et al., 2017), consisting of 10 to 12 different sequences of motions (e.g., jumping jacks, punching, etc.) for each of the 10 human subjects. In the second split, we divide the data by a random choice of a *different*

GT    Baseline    Ours    GT    Baseline    Ours

Figure 4: Human body part segmentation.

action sequences for each human. This experiment ensures our method can generalize knowledge of action sequences seen in training from one human subject to other human subjects at test time. Finally, in the third split we choose the *same* sequence of movements (e.g., the one-leg jump sequence) to be removed from the training set and be placed as the test set. The last test evaluates the effect of the piecewise $E(3)$ prior, as implemented in our method, to generalize to unseen movement.

In Tab. 1, we report the mean IoU(%) score for all 3 tests. As baseline models, we opt for PointNet (Qi et al., 2017) and DGCNN (Wang et al., 2019) as order invariant point networks. For $E(3)$ invariant networks, our baselines selection includes Vector Neurons (VN) (Deng et al., 2021), VN-Transformer (VN-T) (Assaad et al., 2023), FrameAveraging (FA) (Puny et al., 2022), and Equivariant Point Network (EPN) (Chen et al., 2021) backbone as implemented in the human body part segmentation network de-

| Method | random | unseen random seq. | unseen seq. |
|---|---|---|---|
| PointNet | 84.4 | 78.5 | 80.1 |
| DGCNN | 82.2 | 70.3 | 79.5 |
| VN | 42.4 | 24.8 | 33.3 |
| VN-T | 63.5 | 50.9 | 50.0 |
| FA | 83.5 | 78.1 | 76.7 |
| EPN | 89.6 | 77.8 | 84.1 |
| Ours | **94.2** | **92.2** | **93.5** |

Table 1: Mean IoU(%) test set score for human body parts segmentation.

scribed in Feng et al. (2023). Fig. 4 shows qualitative test results of an unseen random seq. pose (first row) and an unseen random pose (second row). We conclude from the results that (i) our framework is a valid backbone with similar expressive power as common point network baselines, (ii) our framework utilizes piecewise $E(3)$ equivariance to gain better generalization across human subjects than baseline approaches and, (iii) piecewise $E(3)$ equivariant prior can help to generalize to unseen movements.

Lastly, to test the versatility of our framework, we evaluate it on a point cloud classification task. On that hand, we consider the DFaust subset of AMASS (Mahmood et al., 2019), consisting of 9 human subjects. We define the task of classifying a model to a subject. For testing, we use an "out of distribution" test set from PosePrior (Akhter & Black, 2015). The results from this experiment support the usability of our framework for classification tasks as well. The detailed report can be found in the Appendix, including all the hyper-parameters used for the experiments in this section.

## 3.2 ROOM SCANS

In this section, we test the potential of our framework for *one-shot* generalization. To that end, we employ a dataset of 8 scenes capturing a real-world room where the furniture in the room has been positioned differently in each of the 8 scans for each scene. Within each scan, there are 3 to 4 labeled furniture-type objects, including the floor. The task objective is to assign each input point to one of the object instances composing the scene. In addition to the difficulty of segmenting moving objects in the scene, solutions to this task must handle noise and sampling artifacts arising from the scanning procedure. For instance, scans of objects occasionally contain holes or exhibit ghost geometry. Here

we compare two alternative solutions this this task: (1) we only train our method using a *single* scan, and test its generalization to the other seven scans of the same scene. (2) We train baseline networks on the large-scale synthetic shape segmentation dataset from Huang et al. (2021a), which randomly samples independent motions for multiple objects taken from ShapeNet (Chang et al., 2015).

In Tab. 2 we report the mean IoU(%) test score for each of the scenes. Fig. 3 shows qualitative results for 2 rooms. Despite only training on a single scan, our model outperforms baselines trained on a large synthetic dataset in 7 out of the 8 test scenes. These results suggest potential advantages of using piecewise $E(3)$ equivariant architectures in a single shot setting over the use of large-scale synthetic data. Furthermore, to make baseline approaches work, we employed a RANSAC algorithm to identify the ground plane, with an inlier distance threshold of 0.02 and 1000 RANSAC iterations. In contrast, our method requires *no* preprocessing since the network can treat the floor as it would for any other part of the input data.

| Method | Scene 1 | Scene 2 | Scene 3 | Scene 4 | Scene 5 | Scene 6 | Scene 7 | Scene 8 |
|---|---|---|---|---|---|---|---|---|
| PointNet | $33.0 \pm 8.5$ | $50.2 \pm 4.3$ | $31.1 \pm 3.4$ | $38.3 \pm 4.3$ | $36.7 \pm 6.1$ | $45.2 \pm 22.0$ | $57.4 \pm 1.5$ | $36.6 \pm 4.6$ |
| DGCNN | $36.7 \pm 3.6$ | $38.8 \pm 10.8$ | $41.8 \pm 4.9$ | $31.0 \pm 2.7$ | $48.9 \pm 4.3$ | $35.1 \pm 8.4$ | $59.5 \pm 7.3$ | $35.4 \pm 6.3$ |
| VN | $13.0 \pm 2.8$ | $18.6 \pm 1.5$ | $24.7 \pm 0.8$ | $15.2 \pm 1.1$ | $24.4 \pm 1.1$ | $17.6 \pm 1.7$ | $25.6 \pm 1.0$ | $23.0 \pm 1.2$ |
| Ours | $\textbf{88.0} \pm 13.0$ | $\textbf{98.2} \pm 0.7$ | $\textbf{97.4} \pm 1.5$ | $\textbf{96.3} \pm 2.0$ | $\textbf{93.2} \pm 3.9$ | $93.4 \pm 2.8$ | $\textbf{83.3} \pm 13.3$ | $\textbf{92.2} \pm 1.8$ |
| PointNet (Synthteic) | $76.6 \pm 22.4$ | $97.3 \pm 2.1$ | $91.2 \pm 4.8$ | $89.7 \pm 4.0$ | $91.9 \pm 5.1$ | $95.1 \pm 1.0$ | $66.6 \pm 9.7$ | $83.2 \pm 4.0$ |
| DGCNN (Synthetic) | $77.5 \pm 22.3$ | $93.7 \pm 10.9$ | $97.1 \pm 0.7$ | $84.4 \pm 13.0$ | $89.1 \pm 16.6$ | $\textbf{95.6} \pm 1.1$ | $76.2 \pm 10.6$ | $90.6 \pm 6.2$ |
| VN (Synthteic) | $65.5 \pm 18.7$ | $93.7 \pm 4.9$ | $80.7 \pm 17.6$ | $59.3 \pm 11.0$ | $92.5 \pm 4.9$ | $82.5 \pm 15.0$ | $77.4 \pm 6.1$ | $62.0 \pm 12.9$ |

Table 2: One-shot generalization on real-world scans from the Dynlab dataset (Huang et al., 2021a).

## 4 RELATED WORK

**Global Equivariance.** We introduce a novel method for piecewise $E(3)$ equivariance in point networks. Euclidean group symmetry has been studied in point networks mainly in describing architectures that accommodate global transformations (Chen et al., 2019; Thomas et al., 2018; Fuchs et al., 2020; Chen et al., 2021; Deng et al., 2021; Assaad et al., 2023; Zisling & Sharf, 2022; Katzir et al., 2022; Poulenard & Guibas, 2021; Puny et al., 2022). These was shown to perform well in various applications including reconstruction (Deng et al., 2021; Chatzipantazis et al., 2022; Chen et al., 2022), pose estimation (Li et al., 2021; Lin et al., 2023; Pan et al., 2022; Sajnani et al., 2022; Zhu et al., 2022), and robot manipulation (Simeonov et al., 2022; Higuera et al., 2023; Xue et al., 2023) tasks. Some works have dealt with respecting the symmetry by manipulating their input representation (Deng et al., 2018; Zhang et al., 2019; Gojcic et al., 2019). A popular line of work utilizes the theory of spherical harmonics to achieve equivariance (Worrall et al., 2017; Esteves et al., 2018; Liu et al., 2018; Weiler et al., 2018; Cohen et al., 2018).

**Object-Level and Part-Based Equivariance** Several works have studied the equivariance of parts. EON (Yu et al., 2022) and EFEM (Lei et al., 2023) both studied object-level equivariance in scenes. EON used a manually tuned 'suspension' to compute an equivariant object frame in which the context is aggregated. In EFEM, instance segmentation is achieved by training a shape prior using a shape collection, and employing it to refine scene regions. Instead, we do not assume prior knowledge of the underlying partition. Equivariance for per-part pose estimation in articulated shape was devised in Liu et al. (2023). Yet their self-supervised approach relies on part grouping according to features that are invariant to *global* rotations which may result in unknown errors when local transformations are introduced. Part-based equivariance was also studied for segmentation in Deng et al. (2023), relying on an intriguing fixed-point convergence procedure.

## 5 CONCLUSION

We presented APEN, a point network design for approximately piecewise $E(3)$ equivariant models. We implemented APEN networks to tackle recognition tasks such as point cloud segmentation, and classification, demonstrating superior generalization over common baselines. On the theoretical side, our work lays the ground for an analysis of piecewise equivariant networks in terms of their equivariance approximation error. The bounds we present in this study serve as merely initial insights on the possibility of controlling the equivariance approximation error, and further analysis of our suggested bounds is marked as an interesting future work. Further extending this framework for other 3D tasks, e.g., generative modeling and reconstruction is another interesting research venue.

ACKNOWLEDGMENTS

The authors would like to thank Jonah Philion for the insightful discussions and valuable comments. Or Litany is a Taub fellow and is supported by the Azrieli Foundation Early Career Faculty Fellowship.

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

## A  APPENDIX

### A.1  PROOFS

#### A.1.1  PROOF OF LEMMA 1

*Proof.* (Lemma 1) Let $\boldsymbol{X} \in U$, $\boldsymbol{Z} \in \{0, 1\}^{n \times k}$, and $g \in G$. Then,

$$\psi(g \cdot (\boldsymbol{X}, \boldsymbol{Z}), \boldsymbol{Z}) = \sum_{j=1}^{k} \psi_b(g \cdot (\boldsymbol{X}, \boldsymbol{Z}) \odot \boldsymbol{Z}\boldsymbol{e}_j \mathbf{1}_d^T) \odot \boldsymbol{Z}\boldsymbol{e}_j \mathbf{1}^T =$$

$$\sum_{j=1}^{k} \psi_b(\sum_{j=1}^{k} (g_j \cdot \boldsymbol{X}) \odot (\boldsymbol{Z}\boldsymbol{e}_j \mathbf{1}_d^T)) \odot \boldsymbol{Z}\boldsymbol{e}_j \mathbf{1}^T = \sum_{j=1}^{k} g_j \cdot \psi_b(\boldsymbol{X} \odot \boldsymbol{Z}\boldsymbol{e}_j \mathbf{1}_d^T) \odot \boldsymbol{Z}\boldsymbol{e}_j \mathbf{1}^T$$

where the last equality follows from the fact the $\psi_b$ is $E(d)$ equivariant and the second equality from the fact that $\boldsymbol{Z}\boldsymbol{e}_j \odot \boldsymbol{Z}\boldsymbol{e}'_j = \mathbf{0}$ for $j \neq j'$. Lastly, for any permutation $\sigma_k(\cdot)$, we have,

$$\sum_{j=1}^{k} \psi_b(\boldsymbol{X} \odot \boldsymbol{Z}\boldsymbol{e}_{\sigma_k(j)} \mathbf{1}_d^T) \odot \boldsymbol{Z}\boldsymbol{e}_{\sigma_k(j)} \mathbf{1}^T = \sum_{j=1}^{k} \psi_b(\boldsymbol{X} \odot \boldsymbol{Z}\boldsymbol{e}_{j)} \mathbf{1}_d^T) \odot \boldsymbol{Z}\boldsymbol{e}_j \mathbf{1}^T$$

$\square$

### A.1.2 PROOF OF THEOREM 1

*Proof.* (Theorem 1) Let $\phi : U \to U'$ be of the form

$$\phi(\boldsymbol{X}) = \sum_{j=1}^{k} \psi_b(\boldsymbol{X} \odot \boldsymbol{Z}_* \boldsymbol{e}_j \boldsymbol{1}_d^T) \odot \boldsymbol{Z}_* \boldsymbol{e}_j \boldsymbol{1}^T, \tag{12}$$

where $(\boldsymbol{Z}_*)_{i,:} = \boldsymbol{e}_{\arg\max_j Q(\boldsymbol{Z}|\boldsymbol{X})_{ij}}$, and $\psi_b : U \to U'$ is an $E(d)$ equivariant backbone.

Let $A = \{\boldsymbol{Z} \neq \boldsymbol{Z}_*\}$. Then,

$$Q(A) \leq \sum_{i=1}^{n} (1 - Q(\boldsymbol{e}_i \boldsymbol{Z} = \boldsymbol{e}_i \boldsymbol{Z}_*)) = \sum_{i=1}^{n} (1 - Q_{ij(i)_*})$$

where $j(i)_* = \arg\max_j Q_{ij}$. Then, we set

$$\delta(Q) = \sum_{i=1}^{n} (1 - Q_{ij(i)_*}).$$

Clearly $\delta$ satisfies conditon 3. Now, Let $Q$ satisfying condition 4 w.r.t. $\lambda$. Let $B = \left\{ \boldsymbol{Z} \mid \exists\, 1 \leq i, j \leq n \text{ s.t. } \left(\boldsymbol{Z}\boldsymbol{Z}^T\right)_{ij} > \left(\widehat{\boldsymbol{Z}}\widehat{\boldsymbol{Z}}^T\right)_{ij} \right\}$. Then,

$$\{\boldsymbol{Z}\} = (B \cap A) \cup (B \cap A^C) \cup (B^C \cap A) \cup (B^C \cap A^C).$$

Note that for $B^C \cap A^C$ there is no equivariance approximation error. For $(B \cap A)$, and $(B^C \cap A)$ we can bound using $\delta(Q)$. Lastly, $\boldsymbol{Z} \in (B \cap A^C)$ means $\boldsymbol{Z}_*$ is a "bad" partition, thus $\lambda(Q) \leq \lambda(Q_{\text{simple}})$. To conclude, we use a union bound composed of the decomposition above to get that,

$$\mathbb{E}_{Q_{\boldsymbol{Z}|\boldsymbol{X}}} \|\phi(g \cdot (\boldsymbol{X}, \boldsymbol{Z})) - g \cdot (\phi(\boldsymbol{X}), \boldsymbol{Z})\| \leq (\lambda(Q_{\text{simple}}) + \delta(Q))M.$$

$\square$

### A.2 Q PREDICTION

In this section we provide an empirical validation to the expected behavior of $\lambda(Q_{\text{simple}})$ as $k \to n$. To that end, we examine a 2D toy example, featuring $n = 14$ points partitioned to 3 groups. Figure 5 shows this toy example, with distinct colors denoting the ground truth partition. Figure 6 shows a plot of $\lambda(Q)$ values for $k \in [1, 14]$. The green line shows $\lambda(Q)$ for the simple $Q$ model, defined by a uniform draw of $k$ parts partition, where each part includes at least one point. The red line shows $\lambda(Q)$ for a $Q$ model, defined by a Voronoi partition with centers drawn randomly proportionally to $k$ furthest point sampling. Note that as expected, $\lambda(Q) \to 0$ as $k \to n$.

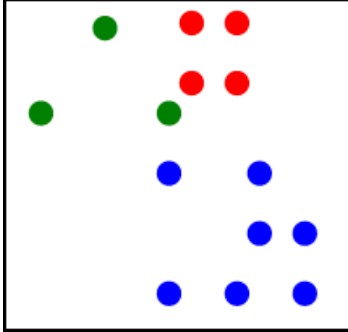

Figure 5: 2D toy example consisting of $n = 14$ points, partitioned into 3 parts.

Next, we provide in Alg. 1 a detailed description of our $Q$ prediction algorithm.

### A.3 ADDITIONAL IMPLEMENTATION DETAILS

#### A.3.1 ARCHITECTURE

We start by describing our concrete construction for the encoder, e and d used in our experiments. The network consists of APEN layers of the form,

$$\text{APEN}(n, a_{\text{in}}, b_{\text{in}}, a_{\text{out}}, b_{\text{out}}) : \mathbb{R}^{n \times (a_{\text{in}} + 3 \times b_{\text{in}})} \to \mathbb{R}^{n \times (a_{\text{out}} + 3 \times b_{\text{out}})}$$

Then, the encoder consists of the following blocks:

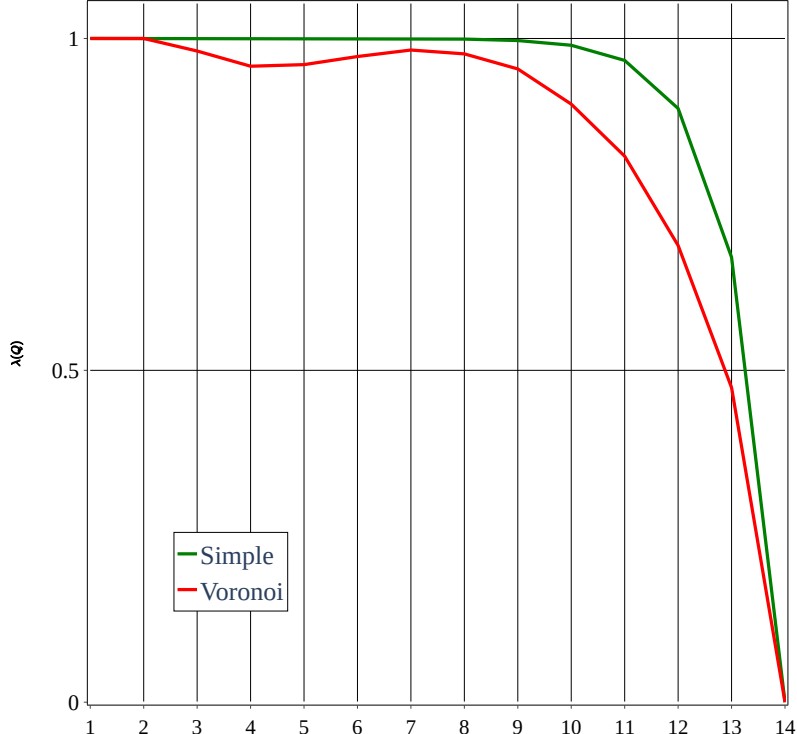

Figure 6: The probability of drawing a bad partition, $\overset{k}{\lambda}(Q_{\text{simple}})$, as $k \to n$, for a 2D toy example with $n = 14$ points.

$$\text{APEN}(n, 0, 2, 17, 5) \to$$
$$\text{APEN}(n, 17, 5, 17, 5) \to$$
$$\text{APEN}(n, 0, 2, 17, 5) \to \text{APEN}(n, 0, 2, 65, 21).$$

The decoder consists of the following block for the segmentation task:

$$\text{APEN}(n, 65, 21, 24, 0),$$

and for the classification task:

$$\text{APEN}(1, 65, 21, 9, 0).$$

Each APEN block is built on equivariant backbone, implemented with Frame Averaging. In turn, the backbone symmetrize a pointnet network $\psi$. We now describe its details.

The network consists of layers of the form

$$\text{FC}(n, d_{\text{in}}, d_{\text{out}}) : \boldsymbol{X} \mapsto \nu\left(\boldsymbol{X}\boldsymbol{W} + \mathbf{1}\boldsymbol{b}^T\right)$$
$$\text{MaxPool}(n, d_{\text{in}}) : \boldsymbol{X} \mapsto \mathbf{1}[\max \boldsymbol{X}\boldsymbol{e}_i]$$

where $\boldsymbol{X} \in \mathbb{R}^{n \times d_{\text{in}}}$, $\boldsymbol{W} \in \mathbb{R}^{d_{\text{in}} \times d_{\text{out}}}$, $\boldsymbol{b} \in \mathbb{R}^{d_{\text{out}}}$ are the learnable parameters, $\mathbf{1} \in \mathbb{R}^n$ is the vector of all ones, $[\cdot]$ is the concatenation operator, $\boldsymbol{e}_i$ is the standard basis in $\mathbb{R}^{d_{\text{in}}}$, and $\nu$ is the ReLU activation. We used the following architecture for the first APEN layer:

$$\text{FC}(n, 6, 96) \overset{L_1}{\to} \text{FC}(n, 96, 128) \overset{L_2}{\to} \text{FC}(n, 128, 160) \overset{L_3}{\to} \text{FC}(n, 160, 192) \overset{L_4}{\to}$$

$$\text{FC}(n, 192, 224) \overset{L_5}{\to} \text{MaxPool}(n, 224) \overset{L_6}{\to} [L_1, L_2, L_3, L_4, L_5, L_6] \overset{L_7}{\to}$$

$$\text{FC}(n, 1024, 256) \overset{L_8}{\to} \text{FC}(n, 256, 256) \overset{L_9}{\to} \text{FC}(n, 128, 32).$$

---

**Algorithm 1** Q prediction

---

**Input:** $Y$; $\tau > 0$ merge threshold and $f$ merge frequency

$i \leftarrow 0$
$(\boldsymbol{\mu}_j) \leftarrow$ *random furthest point sample of $k$ points from* $Y$
$\pi_j \leftarrow \frac{1}{k}$
**while** $i < $ *max iter* **do**

$\quad \gamma_{ij} \leftarrow \frac{\pi_j \mathcal{N}(Y_i; \boldsymbol{\mu}_j)}{\sum_l \pi_l \mathcal{N}(Y_i; \boldsymbol{\mu}_l)}$
$\quad \boldsymbol{\mu}_j \leftarrow \sum_i \frac{\gamma_{ij}}{\sum_{i'} \gamma_{i'j}} Y_i$
$\quad \pi_j \leftarrow \frac{\sum_i \gamma_{ij}}{n}$
$\quad$ **if** $i \bmod f == 0$ **then**
$\quad\quad (j, j') \leftarrow \underset{\{j,j'\} \in \{j | \pi_j > 0\}}{\arg\min} D_{\mathrm{KL}}(\mathcal{N}(\cdot; \boldsymbol{\mu}_j) \| \mathcal{N}(\cdot; \boldsymbol{\mu}'_j))$
$\quad\quad d \leftarrow D_{\mathrm{KL}}(\mathcal{N}(\cdot; \boldsymbol{\mu}_j) \| \mathcal{N}(\cdot; \boldsymbol{\mu}'_j))$
$\quad\quad$ **while** $d < \tau$ **do**
$\quad\quad\quad \pi_j \leftarrow \pi_j + \pi'_j$
$\quad\quad\quad \pi'_j \leftarrow 0$
$\quad\quad\quad (j, j') \leftarrow \underset{\{j,j'\} \in \{j | \pi_j > 0\}}{\arg\min} D_{\mathrm{KL}}(\mathcal{N}(\cdot; \boldsymbol{\mu}_j) \| \mathcal{N}(\cdot; \boldsymbol{\mu}'_j))$
$\quad\quad\quad d \leftarrow D_{\mathrm{KL}}(\mathcal{N}(\cdot; \boldsymbol{\mu}_j) \| \mathcal{N}(\cdot; \boldsymbol{\mu}'_j))$
$\quad\quad$ **end while**
$\quad$ **end if**
$\quad i \leftarrow i + 1$
**end while**
$(\tilde{\boldsymbol{\mu}}_j, \tilde{\pi}_j) \leftarrow (\boldsymbol{\mu}_j, \pi_j)$
$(\boldsymbol{\mu}^*_j, \pi^*_j) \leftarrow (\tilde{\boldsymbol{\mu}}_j, \tilde{\pi}_j) + I^{-1}(\tilde{\boldsymbol{\mu}}_j, \tilde{\pi}_j) s(Y; (\tilde{\boldsymbol{\mu}}_j, \tilde{\pi}_j))$
$Q^{\mathrm{pred}}_{ij} \leftarrow \frac{\mathcal{N}(y_i; \boldsymbol{\mu}^*_j, \sigma) \pi^*_j}{\sum_{j=1}^k \mathcal{N}(y_i; \boldsymbol{\mu}^*_j, \sigma) \pi^*_j}$

**Output**: $Q^{\mathrm{pred}}$, a (differential) minimizer of $E(Y)$

---

For the second and third,

$$\mathrm{FC}(n, 32, 96) \overset{L_1}{\to} \mathrm{FC}(n, 96, 128) \overset{L_2}{\to} \mathrm{FC}(n, 128, 160) \overset{L_3}{\to} \mathrm{FC}(n, 160, 192) \overset{L_4}{\to}$$

$$\mathrm{FC}(n, 192, 224) \overset{L_5}{\to} \mathrm{MaxPool}(n, 224) \overset{L_6}{\to} [L_1, L_2, L_3, L_4, L_5, L_6] \overset{L_7}{\to}$$

$$\mathrm{FC}(n, 1024, 256) \overset{L_8}{\to} \mathrm{FC}(n, 256, 256) \overset{L_9}{\to} \mathrm{FC}(n, 128, 32).$$

And lastly,

$$\mathrm{FC}(n, 32, 96) \overset{L_1}{\to} \mathrm{FC}(n, 96, 128) \overset{L_2}{\to} \mathrm{FC}(n, 128, 160) \overset{L_3}{\to} \mathrm{FC}(n, 160, 192) \overset{L_4}{\to}$$

$$\mathrm{FC}(n, 192, 224) \overset{L_5}{\to} \mathrm{MaxPool}(n, 224) \overset{L_6}{\to} [L_1, L_2, L_3, L_4, L_5, L_6] \overset{L_7}{\to}$$

$$\mathrm{FC}(n, 1024, 256) \overset{L_8}{\to} \mathrm{FC}(n, 256, 256) \overset{L_9}{\to} \mathrm{FC}(n, 128, 128).$$

### A.3.2 HYPER PARAMETERS AND TRAINING DETAILS

We set $\sigma_l = (0.002, 0.005, 0.008, 0.1)$. The number of iterations for the EM was 16. We trained our networks using the ADAM (Kingma & Ba, 2014) optimizer, setting the batch size to 8. We set a fixed learning rate of 0.001. All models were trained for 3000 epochs. Training was done on a single Nvidia V-100 GPU, using PYTORCH deep learning framework (Paszke et al., 2019).

### A.4 ADDITIONAL RESULTS

In this section, we present visualizations of the learned partitions $Q^{\mathrm{pred}}$ across layers in the APEN encoder. Figure 7 shows the learned APEN encoder layers partitions from the experiment in section

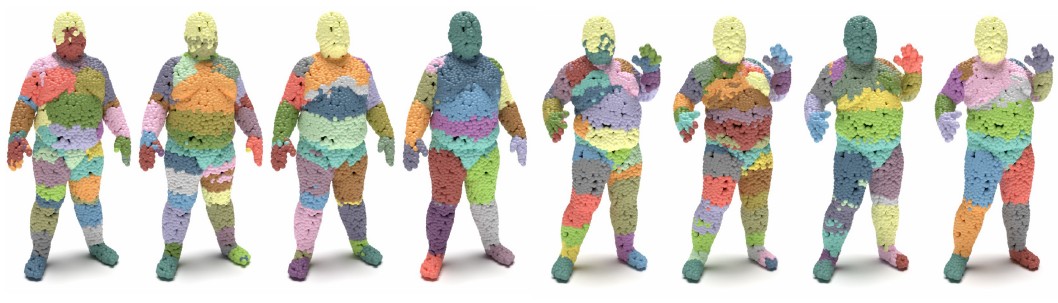

Figure 7: APEN encoder's learned partitions, $Q^{\text{pred}}$, extracted from two test-set examples in the human body segmentation experiment. In each group of 4 elements, the leftmost column shows $Q^{\text{pred}}$ partitions, with subsequent layers' partitions ordered left-to-right, culminating in the rightmost column that shows the encoder's last layer partition.

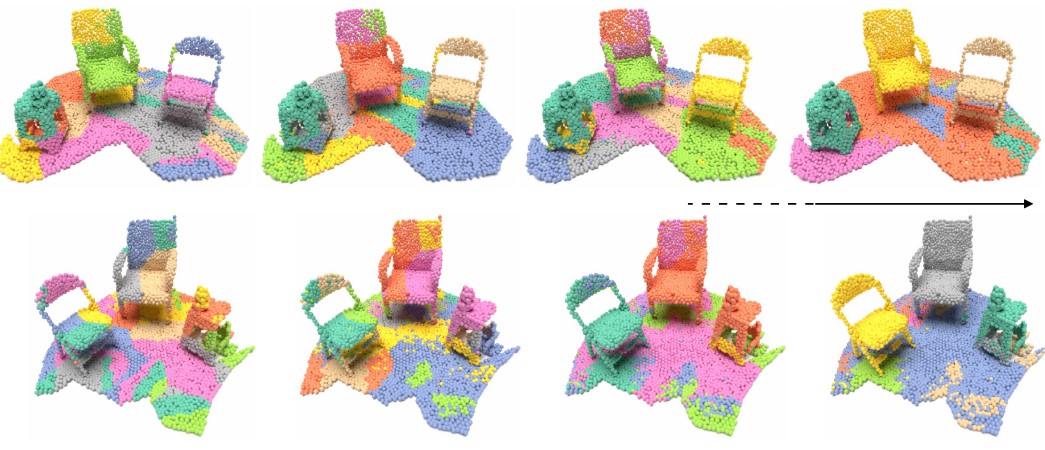

Figure 8: APEN encoder's learned partitions, $Q^{\text{pred}}$, extracted from the one shot segmentation experiment. In the top row, layer partitions of a single training example are shown, while the bottom row shows layer partitions of an unseen test example. The leftmost column shows $Q^{\text{pred}}$ partitions, with subsequent layers' partitions ordered left-to-right, culminating in the rightmost column that shows the encoder's last layer partition.

3.1, while Figure 8 shows partitions from the experiment in section 3.2. Each input point is assigned distinctive colors according to $\arg\max_j Q_{ij}^{\text{pred}}$. It is worth noting that progressing from left to right, the predicted partitions tend to become coarser, a behavior encouraged by setting the hyper-parameter $\sigma_{l+1} > \sigma_l$.

## A.5 SUBJECT CLASSIFICATION EXPERIMENT

| Method | PointNet | DGCNN | VN | Ours |
|---|---|---|---|---|
| Accuracy (%) | 18.5 | 32.1 | 28.2 | 71.4 |

Table 3: Subject classification accuracy comparison.

Here we provide the results of the point cloud classification experiment described in the main text. Fig. 9 shows several typical examples from the considered split. Note the relatively large difference in the distribution of poses. Tab. 3 logs the quantitative evaluation, validating our framework's superiority in this case as well.

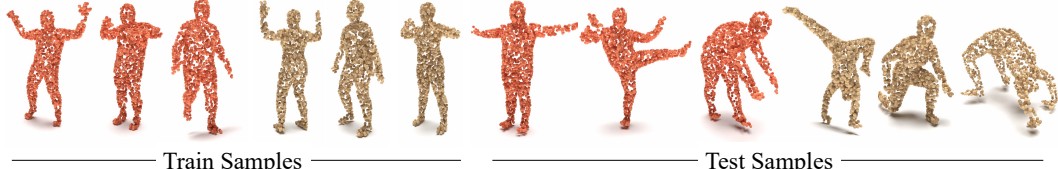

Train Samples ———————— ———————— Test Samples

Figure 9: Training and test set visualization for the subject classification task.

