# OpenReview forum: "Approximately Piecewise E(3) Equivariant Point Networks"
_ICLR.cc/2024/Conference — ICLR 2024 poster_

### Official Review · Reviewer_vmat · 2023-11-01

**Soundness:** 3 good
**Presentation:** 2 fair
**Contribution:** 3 good
**Rating:** 6
**Confidence:** 2

**Summary:**

The paper proposes a new neural network design that approximates piecewise E(3) equivariance. This design allows to segment better deformable or movable objects such as human body parts or objects in a 3D scene.

**Strengths:**

I believe this paper tackles a relevant problem, piecewise network equivariance. A solution for this problem will partially solve some open problems in 3D scene understanding such as 3D scan segmentation or instance segmentation by design of the network.

Moreover, the solution proposed is simple and easy to implement: Initialize the network with an initial partition, use E(3) equivariant layers to process the parts, and then use a clustering algorithm to determine the new set of parts from the predicted features.

**Weaknesses:**

Although I liked the paper and the proposed solution, I believe the paper was very difficult to follow and needs an improved evaluation. In the following paragraphs, I listed some of the things that were unclear to me, maybe I did not fully understand these parts, and some points on how to improve the evaluation of the paper:

- It was not clear to me the statements at the end of section 2.3. How $\sigma \rightarrow 0$ makes $\lambda (Q^{pred}) \rightarrow \lambda (Q^{simple})$ ? There is no control over the features that come out from the previous layers and the clustering obtained can be worse than $Q^{simple}$ as defined later in the paper, farthest point sampling with KNN. I suppose here $Q^{simple}$ refers to the random uniform partition and not the Voronoi partition used later. If this is true, later I would not refer to the Voronoi partition as $Q^{simple}$.

- Moreover, how sigma is defined to achieve the desired number of parts (number of group truth parts)? How robust is the proposed method to different choices of $\sigma$?

- Maybe I missed it, but I think $\delta$ is never properly defined. Moreover, Fig.1 is not helpful unless more context is given. What Fig. 1 illustrates?

- In the Training details paragraph, $Y$ is used to refer to part centers but in the Q Prediction paragraph is used to refer to per-point features. In this same paragraph, the centers are referred as $\mu$ instead. Moreover, $\pi$ is never defined. From the formulas, I suppose it refers to the per-point weights of each of the Gaussians in the GMM.

- In Training details, it is not clear how the loss would work. $Y_{GT} \in R^{n \times d}$, but $Y_l \in R^{k \times d}$ if they are the part centers as the text indicates. Moreover, is each layer supervised by this loss or only the last one?

- In network architecture, it is not clear which network architecture is used. First, it indicates that PoinNet is used but later SpareConv layers are used. From the appendix, I believe all layers are PointNet layers, but I don't understand where the SparseConv layers come into play.

- Regarding evaluation, I believe it needs significant improvement before publication. Here is a list of possible experiments that would help improve the evaluation:
	- Comparison with different choices of sigma in the layers.
	- Visualization of the intermediate partition assignments will help understand the behavior of the model.
	- Experiments on other datasets, such as ShapeNet part segmentation and real 3D scans such as ScanNet.
	- Several relevant baselines missing. Only one global rotation equivariant network is used in the comparisons, Vector Neurons. It would be necessary to compare to network architectures that are locally equivariant such as E2PN, LieConv, or simple graph convolution networks. It is not relevant to show improvement over basic point architectures such as PointNet or DGCNN.

**Questions:**

See weaknesses.

---

> ### Author Response · Authors · 2023-11-15
>
> We Thank the reviewer for a detailed review. Below we address the main comments and questions expressed in this review.
>
> **Q:** It was not clear to me the statements at the end of section 2.3. How $\sigma \rightarrow 0$ makes $\lambda(Q^{\text{pred}}) \rightarrow \lambda(Q^{\text{simple}})$. There is no control over the features that come out from the previous layers and the clustering obtained can be worse.
>
> **A:**  When $\sigma \rightarrow 0$, any arbitrary choice of center points ($\mu_j$) from $Y$, and point assignments ($Q_{ij}$) is a minimizer of the likelihood energy defined in eq. (9). In particular, $Q^{\text{simple}}$ is a minimizer. In the algorithm to compute a minimizer of (9) (see algorithm 1 in the Appendix), the iterative process starts from $Q^{simple}$. The above argument is true for any $Y$ values.
>
> **Q:** How $\sigma$ is defined to achieve the desired number of parts (number of group truth parts)? How robust is the proposed method to different choices of $\sigma$?
>
> **A:** We wish to clarify that in our design, the hyper-parameter $\sigma$ is not suggested to achieve the desired number of parts. The original text did not include such a claim about $\sigma$. Following the reviewer’s comment, we will emphasize this fact in the next revision. The primary restriction on the parameter $\sigma$ (which is stated in the paper) is that it should be set such that $\sigma_l < \sigma_{l+1}$ where $\sigma_l$ is the corresponding $\sigma$ hyperparameter of the $l$-th layer (See the last paragraph in section 2.3 ). This restriction encourages coarser partition predictions across layers, as wider Gaussian centers can “explain” a wider distribution of part prediction features.  In our experiments we set $ (\sigma_l) = (0.002,0.005,0.008,0.1)$ (See Supplementary). We used the same values of $\sigma$ both to the humans and furniture type models experiments, suggesting that $\sigma$ is not sensitive to input types. We note that $\sigma$ has a geometric meaning. For example, one may choose the maximal $\sigma$ with respect to the minimal distance between ground-truth part centers to prevent the predicted partition from becoming coarser than the true one.
>
> **Q:** Maybe I missed it, but I think $\delta$ is never properly defined. Moreover, Fig.1 is not helpful unless more context is given. What Fig. 1 illustrates?
>
> **A:**  $\delta$ measures how much $Q$ is close to a degenerate distribution, i.e. a draw of one and only one partition. This is defined in eq (3). Figure 1 illustrates this property of $\delta$: the triangle depicts all possible $Q$; the vertices illustrate degenerate distributions, while the colors depict the values of the measure $\delta$. We made the formulation of $\delta$ general, as one can come up with different measures $\delta$ satisfying eq. (3), perhaps leading to other models satisfying definition 1. In theorem 1, we show how our construction satisfies definition 1, and choose a specific $\delta$ satisfying eq. (3) and eq. (4) (See proof in Supplementary).
>
> **Q:** In the Training details paragraph, $Y$ is used to refer to part centers but in the Q Prediction paragraph is used to refer to per-point features. In this same paragraph, the centers are referred as $\mu$ instead. Moreover, $\pi$  is never defined.
>
> **A:** $\mu$ and $\pi$ are defined in the last paragraph of the method section, describing the Q prediction. As for $Y$, in the training details and the method section, $Y$ refers to the same thing: per point equivariant prediction of $\phi(X)$.  In the uploaded revision, we added to the training details the dimensions of each element for clarity.
>
> **Q:** In Training details, it is not clear how the loss would work. Moreover, is each layer supervised by this loss or only the last one?
>
> **A:** $Y_l$ in the training details refers to the $l^\text{th}$ layer’s per point equivariant feature of $\phi(X)$ as defined in the method section. In the revision, we clarified this in the training details. We suggested supervising $Y_l$ for all layers.
>
> **Q:** In network architecture, it is not clear which network architecture is used. First, it indicates that PointNet is used but later SpareConv layers are used. I don't understand where the SparseConv layers come into play.
>
> **A:** Our architecture employs a shared $E(3)$ equivariant backbone across parts. Indeed, the shared equivariant backbone is based on PointNet construction. That is, a siamese MLP (between points) and global pooling. The main implementation challenge our framework tackles is related to the fact that the same backbone needs to be applied to different parts where each part has a different number of points (see Figure 2). Therefore, it will be memory-efficient to represent this input as a sparse tensor. We used sparse linear layers from Choy et al. to implement the MLP in our pointnet backbone. Lastly, we assure the reviewer that we are committed to publishing our code upon publication.

---

> > ### Author Response · Authors · 2023-11-15
> >
> > **Q:** Several relevant baselines missing. It would be necessary to compare to network architectures that are locally equivariant such as E2PN, LieConv, or simple graph convolution networks. It is not relevant to show improvement over basic point architectures such as PointNet or DGCNN.
> >
> > **A:** We respectfully disagree with some of the claims above. First, a simpler alternative approach to equivariance networks is data augmentation of non-equivariant methods such as PointNet and DGCNN. The first split in the human experiment tests this alternative. Furthermore, the tasks considered in this work are piecewise equivariant, thus it is not reasonable to assume global equivariant methods should perform better than non-equivariant methods. Second, it is not clear what the reviewer means by simple graph convolutions, as the focus of this paper is point clouds and not graphs. In case the reviewer is referring to a network extracting features from local point neighborhoods, DGCNN and VectorNeurons (VN) (with the DGCNN backbone) do exactly this.  Lastly, it is not clear what the reviewer means by stating the above baselines as local equivariant. For example, E2PN, similarly to VN, calculates features from points local neighborhood but these features are only global SO(3) equivariant as they are defined w.r.t. a single frame of reference.
> > To the best of our knowledge, we are not familiar with general frameworks for local E(3) equivariant point networks. See also the last answer to reviewer gDD2. Nevertheless, we do agree that it would be useful to validate the above claims on more than one equivariant baseline. We plan to incorporate these results in the next revision, hopefully before the end of the discussion period.
> >
> > **Q:** Experiments on other datasets are required, such as ShapeNet part segmentation and real 3D scans such as ScanNet.
> >
> > **A:** We respectfully disagree with some of the claims above. Our work claims that when a piecewise E(3) symmetry is exhibited in the data, then a piecewise E(3) equivariant network generalizes better than networks that do not impose this prior. We make no claim about our network for cases that do not fit piecewise symmetry. Most of ShapeNet objects are aligned and do not exhibit articulated motion. Thus, we do not consider it a valuable experiment in the scope of this work. As for ScanNet, we thank the reviewer for this suggestion. We agree with the reviewer that it will be interesting to test our method on real-world scans. Note that In this work, we did experiment with real-world room scans (Dynlab dataset). We mark ScanNet as a future work, including the extension of our framework to object detection tasks on ScanNet.
> >
> > **Q:** Visualization of the intermediate partition assignments will help understand the behavior of the model and comparison with different choices of sigma in the layers.
> >
> > **A:** We thank the reviewer for this great suggestion! We plan to add such a visualization in the next revision, to be uploaded before the end of the current discussion period.

---

> > > ### Comment · Reviewer_vmat · 2023-11-22
> > >
> > > I would like to thank the authors for carefully answering all my questions and trying to improve the readability of the paper. I believe most of my concerns have been addressed.
> > >
> > > However, I believe the statements of the authors wrt. E2PN is not correct. E2PN and LieConv are group convolutions that work in local neighborhoods. By restricting the receptive field of the convolution, these operations become locally equivariant to the group in the local region defined by the receptive field. The work of Feng et al.[1] for example, make use of such operations to achieve part equivariance on the different parts of human poses. Therefore, I believe comparing to these operations is highly relevant for this work.
> > >
> > > [1] Generalizing Neural Human Fitting to Unseen Poses With Articulated SE(3) Equivariance, Feng et al. 2023

---

> ### Author Response · Authors · 2023-11-23
>
> We would like to thank the reviewer for reviewing our initial response and the insightful suggestions.
>
> We would like to highlight key updates incorporated in our latest revision, as recommended by the reviewer:
>
> In Table 1, we included the result of Equivariant Point Network (EPN) [1] backbone as implemented in the human body part segmentation network described in [2]. This baseline utilizes local neighborhoods to extract **local** (convolutional) E(3) equivariant features from which, per-point **invariant** segmentation predictions are made. In addition, Table 1 now also includes two additional E(3) invariant networks:  VN-Transformer [3] and FrameAveraging [4].
> Our method consistently outperforms all of these baselines, thereby reinforcing our assertion regarding the advantages of integrating the **approximately** piecewise E(3) equivariant prior into a network.
>
> Additionally, the appendix now includes visualizations (Figure 7 and Figure 8) depicting APEN encoder learned layer partitions. These visuals illustrate the encouragement of gradually coarser partitions through the condition $\sigma_l < \sigma_{l+1}$.
>
> We are pleased to provide further clarification on any aspect of the revised paper if necessary.
>
> [1] Equivariant Point Network for 3D Point Cloud Analysis, Chen et al. [CVPR 2021].
>
> [2] Generalizing Neural Human Fitting to Unseen Poses With Articulated SE(3) Equivariance, Feng et al. [ICCV 2023].
>
> [3] VN-Transformer: Rotation-Equivariant Attention for Vector Neurons, Assaad et al. [TMLR 2023].
>
> [4] Frame Averaging for Invariant and Equivariant Network Design, Puny et al. [ICLR 2022].

---

> > ### Comment · Reviewer_vmat · 2023-11-23
> >
> > I would like to thank the authors for their efforts in trying to address all my issues and comments. The new comparisons and visualization have improved the paper and now put in perspective the new method wrt existing methods. Therefore, I would increase my score.

---

### Official Review · Reviewer_FhAC · 2023-11-02

**Soundness:** 3 good
**Presentation:** 3 good
**Contribution:** 3 good
**Rating:** 8
**Confidence:** 3

**Summary:**

This paper introduces a general framework called APEN for constructing an approximately piecewise-E(3) equivariant neural network for point clouds. The model is then used for classification and part-segmentation tasks on two datasets: scans of human objects performing various sequences of movements and room scans of furniture-type objects.

**Strengths:**

- Errors that arise when maintaining equivariance of point clouds are unavoidable in practice. The model presented in this paper can be employed to control these unavoidable errors. To the best of my knowledge, there have been very few works in the literature that can address this problem. This model can also be seen as a generalization of classical global E(3)-equivariant point networks when the error is set to zero.
- The presented model is constructed in such a way that it can ensure equivariance of partitions of the point clouds, which is more challenging than achieving global equivariance.
- The paper includes a detailed theoretical analysis with useful intuitive explanations.
- The model performs well in the two experiments.

In general, I believe this is a noteworthy paper.

**Weaknesses:**

- The theoretical explanations in the paper are somewhat challenging to comprehend. For example, why the indicator $\lambda(Q)$ in Eq. (2) can be used to measure the probability of drawing a bad partition from a non-proper subpartition of $Z_*$. The rationale behind defining $Q_{simple}$ in that manner and the behavior of the number $\lambda(Q_{simple})$ as $k$ tends to $n$ also need further clarification.
- The approximation error defined in Definition 1 depends on $M$ which is an unknown and possibly large number. Therefore, it is unclear how this approximation error can be employed to control the errors occurring in experiments. For example, given a positive number $\epsilon$ how can we design the model in a way that the approximation error does not exceed $\epsilon$?
- PointNet and DGCNN are non-equivariant models. Therefore, it may not be fair to compare the proposed model with them. Instead, it would be interesting to assess the efficiency of the proposed model compared to other equivariant models for point clouds, with suitable modifications to make them equivariant for partitions of the point clouds.

**Questions:**

See Weaknesses.

---

> ### Author Response · Authors · 2023-11-15
>
> We thank the reviewer for a detailed review. Below we address the main comments and questions expressed in this review.
>
> **Q:** Why Eq. (2) can be used to measure the probability of drawing a bad partition?
>
> **A:** Thank you for taking the time to spot this. For drawing a bad partition it is enough that two points from mixed parts in the ground truth partition $Z_*$ are drawn to be in some shared part in $Z$. This fix for eq. (2) is included in the uploaded revision.
>
> **Q**: The rationale behind defining $Q_{\text{simple}}$ in that manner and the behavior of the number $\lambda(Q_{\text{simple}})$ as $k$ tends to $n$ need further clarification.
>
> **A**: Thank you for this suggestion. Our last revision includes an additional explanation in the method section detailing why the expected behavior of $\lambda(Q_{\text{simple}}) \rightarrow 0$ when $k \rightarrow n$. For the reviewer’s convenience, we include this explanation here as well. To understand $\lambda(Q_{\text{simple}})$ dependency on $k$, consider the corresponding random sequential process to generate a random $k$ parts partition. Clearly, larger values of $k$ result in each part potentially containing fewer points. Since the probability of drawing the next point from mixed ground-truth parts is independent of $k$, determined solely by the number of input points and the ground-truth partition, the probability that the next point generated a bad part lowers as $k$ increases. In the next revision, in the Appendix, we will plot an empirical analysis of $\lambda(Q_{\text{simple}})$ as a function of $k$.
>
> **Q:** The approximation error defined in Definition 1 depends on $M$ which is an unknown and possibly large number. Therefore, it is unclear how this approximation error can be employed to control the errors. For example, given a positive number
> $\varepsilon$, how can we design the model in a way that the approximation error does not exceed $\varepsilon$?
>
> **A:** Thank you for this important observation. Restricting a neural network to satisfy a global bound, i.e. $\sup||f(x)||\leq M$, is a common ingredient in a network design. Often implemented by adding a $\text{tanh}$ or $\text{softmax}$ activation in the last layer. Thus, it is reasonable to assume the network is bounded. The bound $M$ can be selected according to the task at hand. Theoretically, to achieve a required bound $\varepsilon$, one needs to choose $k$ large enough and/or $\sigma$ small enough s.t. $(\lambda(Q_{\text{simple}}) + \delta(Q)) \leq \frac{\varepsilon}{M}$. In the current work, we only claim such a $k$ and $\sigma$ exist. In the paper, we marked that the further analysis required for achieving accurate bounds is out of the current scope, and suggested it as an important future work (see Conclusion section, and the first paragraph in the implementation details).
>
> **Q:** PointNet and DGCNN are non-equivariant models. Therefore, it may not be fair to compare the proposed model with them. Instead, it would be interesting to assess the efficiency of the proposed model compared to other equivariant models for point clouds, with suitable modifications to make them equivariant for partitions of the point clouds.
>
> **A:**  PointNet and DGCNN are popular frameworks for point cloud tasks. Indeed they are not global rotation equivariant. However, the tasks considered in this work enjoy piecewise E(3) equivariant symmetry. Thus, one should not expect a global equivariant network to perform better on these tasks than non-equivariant methods. Moreover, it is worthy to compare equivariant methods to non-equivariant methods + data augmentations as the first split test in section 3.1 suggests. Nevertheless, we plan to include additional comparisons to more global equivariance networks in the next revision (expected to be uploaded before the rebuttal deadline). In addition, it is not entirely clear to us what the reviewer suggests in regards to making suitable modifications to make a global equivariant baselines network to be piecewise equivariant. We understood this comment as a question regarding previous designs in the literature for extracting equivariant features from local neighborhoods. Indeed, computing point features from local neighborhoods is common in many frameworks for point networks. In fact, DGCNN and VectorNeurons (VN) incorporate local features as well. However, for VectorNeurons, these local features are all with respect to a *global* frame of reference. Thus, VN local features are not able to equivariantly share feature information between local neighborhoods that are related by a *local* E(3) transformation. To the best of our knowledge, we are not familiar with general frameworks for local E(3) equivariant point networks. It is worth noting works based on Gauge equivariant networks, such as [1] and [2], but these works operate on manifold data.
>
> [1]  Gauge Equivariant Convolutional Networks and the Icosahedral CNN, Cohen et al.
>
> [2] Gauge Equivariant Transformer, He et al.

---

### Official Review · Reviewer_gDD2 · 2023-11-05

**Soundness:** 3 good
**Presentation:** 2 fair
**Contribution:** 3 good
**Rating:** 6
**Confidence:** 3

**Summary:**

This study aims to extend global E(3)-equivariant point networks by introducing support for multiple components, each subject to its local E(3) symmetries. Prior approaches have unbounded errors when their partition predictions diverge from the actual data. To address this, this paper proposes a framework called APEN to construct approximate piecewise-E(3) equivariant point networks, which offers guaranteed bounds on the resulting approximation errors.

In particular, the APEN model starts with a detailed initial partition and gradually merges these partitions into larger segments in successive layers. The efficacy of this method is validated through tests on 3D scans of interior spaces and human movement patterns.

**Strengths:**

- This paper tackles an important problem: modeling local symmetries for point networks. Previous work provided no guarantees on the approximation error when the predicted partitions are inconsistent with the true partitions.

**Weaknesses:**

- The paper is a bit hard to follow in general. I think the introduction should provide more high-level pictures and intuitions without talking about technical details. Also see questions.
- The baselines used are PointNet, DGCNNs, and VectorNeurons, while this work focuses on local symmetries. It would be useful to discuss how previous methods that extend global E(3) point networks to handle local symmetries perform on these tasks. While the authors likely don't need to run experiments for the rebuttal, it would be good to hear their rationale for choosing these particular baselines over other more directly relevant work.

**Questions:**

I think I don't fully understand this work so I can't make a good judgment yet. It would be greatly appreciated if the authors could clarify the following questions:
- Page 2, what exactly is defined as "equivariant approximation error"? What is the relationship between this and general approximation error? The sentence "this simple model enables bounding the equivariance approximation error solely by the probability of drawing a 'bad' partition. Crucially, this bound is independent of any required restriction on the resulting piecewise equivariant model function bounded variation." is difficult to follow in the introduction without more context, before read all the theorems and technical details afterwards.
- Equation 2, as I understand it, as long as the two partitions are inconsistent, | $||ZZ^T - Z_{\ast}Z_{\ast}^T|| > 0$, even if $Z$ is a finer partition of $Z_{\ast}$? Is that correct? If not, can you explain why?
- Theorem 1, is it possible the $\arg\max$ would give a set rather than a single value (i.e. ties)? Would that cause any problems?
- Table 1, can the authors discuss why the performance of vector neurons are so bad, even compared to baselines?

**Details Of Ethics Concerns:**

I have no ethical concerns.

---

> ### Author Response · Authors · 2023-11-15
>
> We thank the reviewer for a detailed review. Below we address the main comments and questions expressed in this review.
>
> **Q:** Page 2, what exactly is defined as "equivariant approximation error"?
>
> **A:** The equivariance approximation error is the error resulting from a function’s inability to satisfy the equivariance constraint $f(g \cdot X) = g \cdot f(X)$ (see definition 1 in the paper). Since the true partition inducing the symmetry $g$ is unknown, but rather predicted by a model, the error $||f(g\cdot X) - g\cdot f(X)||$ is inevitable. In this work, we opt for a network design providing means to control the equivariance approximation error. In the uploaded revision of the paper, we added a few sentences to the introduction clarifying the definition of the equivariant approximation error.
>
> **Q:** What is the relationship between the equivariance approximation error and general approximation error?
>
> **A:** Thank you for this interesting question. The primary assumption of our work is that the ground truth function is a piecewise equivariant function. Therefore, a model function achieving zero general approximation error implies zero equivariance approximation error.
>
> On the other hand, functions satisfying the equivariance constraint $f(g\cdot X)=g\cdot f(X)$, or the more general constraint we suggested in definition 1 for functions satisfying the bounded equivariance approximation error $||f(g\cdot X)-g\cdot f(X)||\leq (\lambda(Q_{\text{simple}}) + \delta(Q))M$, is insufficient to guarantee a low *general* approximation error. There are many functions satisfying the equivariance condition with arbitrary bad general approximation error (e.g., constant functions). To summarize, zero equivariance approximation error is a necessary but not sufficient condition for zero general approximation error.
>
> **Q**: The sentence "this simple model enables bounding the equivariance approximation error solely by… " is difficult to follow in the introduction without more context, beforeall the theorems and technical details afterwards.
>
> **A**: Thank you for spotting this. We agree with the reviewer's comment. We simplified this paragraph from the introduction in the uploaded revision.
>
> **Q**: Equation 2 condition is satisfied even when $Z$ is finer
>
> **A:** Thank you for spotting this. We fixed the statement in eq (2) such that a bad partition is defined by having at least two points from mixed ground-truth parts. This fix is included in the uploaded revised version.
>
> **Q:** Theorem 1, is it possible that $\text{argmax}$ would give a set? Would that cause any problems?
>
> **A:** Ties in Q implies two or more partitions can be arbitrarily selected by the marginalization scheme $Z_{\dagger}$. If indeed the final partition is arbitrarily selected then theorem 1 does not hold. In practice, since $\delta(Q)$ is small, i.e., Q is concentrated near some partition (see eq. (3)), such cases are not possible.
>
> **Q:** Table 1, can the authors discuss why the performance of vector neurons are so bad, even compared to baselines?
>
> **A:** The tasks considered in this work demonstrate piecewise E(3) equivariant symmetry. In fact, in the human experiment (Section 3.1) most of the human models are globally aligned. Therefore, global equivariant models are not expected to perform better than non-equivariant baselines.
> While, in general, one might not expect global equivariant models to perform worse than non-equivariant methods on tasks with piecewise equivariant characteristics, the specific observation of vector neurons underperforming compared to non-equivariant baselines when the symmetry prior is mismatched is not a novel discovery, and can be attributed to limited expressivity and/or poor convergence properties. This has been demonstrated in other instances in the past literature as well. For example, see Table 1, z\z column in the original vector neurons paper (https://arxiv.org/pdf/2104.12229.pdf).
>
> **Q:**  The baselines chosen don't reflect the focus of this work on local symmetries. It would be useful to discuss how previous methods that extend global E(3) point networks to handle local symmetries perform on these tasks.
>
> **A:**  Computing point features from local neighborhoods is common in many frameworks for point networks. In fact, DGCNN and VectorNeurons (VN) incorporate local features as well. However, for VectorNeurons, these local features are all with respect to a *global* frame of reference. Thus, VN local features are not able to equivariantly share feature information between local neighborhoods that are related by a *local* E(3) transformation. To the best of our knowledge, we are not familiar with general frameworks for local E(3) equivariant point networks. It is worth noting works based on Gauge equivariant networks, such as [1] and [2], but these works operate on manifold data.
>
> [1]  Gauge Equivariant Convolutional Networks and the Icosahedral CNN, Cohen et al.
>
> [2] Gauge Equivariant Transformer, He et al.

---

### Author Response · Authors · 2023-11-23
**Rebuttal Response to All Reviewers**

We want to thank the reviewers for their effort in providing constructive feedback.

In response to the insightful suggestions and concerns raised by the reviewers, we have uploaded a **second** revision. While the first revision primarily focused on enhancing the clarity of our presentation based on the feedback provided by the reviewers, this revision mainly addresses the reviewers' comments concerning the selection of equivariant baselines.

In Table 1, we included the result of Equivariant Point Network (EPN) [1] backbone as implemented in the human body part segmentation network described in [2]. This baseline utilizes local neighborhoods to extract **local** (convolutional) E(3) equivariant features from which, per-point **invariant** segmentation predictions are made. In addition, Table 1 now also includes two additional E(3) invariant networks:  VN-Transformer [3] and FrameAveraging [4].

Our method consistently **outperforms** all of these baselines, thereby reinforcing our assertion regarding the advantages of integrating the **approximately** piecewise E(3) equivariant prior into a network.

In addition, this revision also includes the following changes:

**Visualization of APEN Encoder-Learned Layer Partitions:** In response to reviewer vmat's suggestion, the appendix now includes visualizations of APEN encoder-learned layer partitions. These visualizations align with our network design suggestion, illustrating the encouragement of gradually coarser partitions through the condition $\sigma_l < \sigma_{l+1}$.

**Empirical Validation of $\lambda(Q_{\text{simple}})$**: Addressing reviewer FhAC's comment on the behavior of $\lambda(Q_{\text{simple}})$ as $k\rightarrow n$, we have not only augmented the explanation in the method section (done in first revision) but now also introduced an empirical validation in the Appendix. This validation, performed on a 2D toy example, demonstrates that, as expected, $\lambda(Q_{\text{simple}})\rightarrow 0$ when $k\rightarrow n$.

In summary, the review process has been encouraging, with reviewers acknowledging that our work addresses an important and challenging problem (gDD2, FhAC02, vmat). They appreciate the simplicity of our proposed solution (vmat), particularly as the first one aiming to control the piecewise equivariance error (gDD2), and deem the work noteworthy (FhAC02). We sincerely value the thoughtful review process and believe that addressing major concerns and comments has resulted in an improved paper.

[1] Equivariant Point Network for 3D Point Cloud Analysis, Chen et al. [CVPR 2021].

[2] Generalizing Neural Human Fitting to Unseen Poses With Articulated SE(3) Equivariance, Feng et al. [ICCV 2023].

[3] VN-Transformer: Rotation-Equivariant Attention for Vector Neurons, Assaad et al. [TMLR 2023].

[4] Frame Averaging for Invariant and Equivariant Network Design, Puny et al. [ICLR 2022].

---

### Meta-Review · Area_Chair_Rom9 · 2023-12-09

**Metareview:**

This work studies approximate equivariance, which is useful in many areas such as physics, point cloud etc. The work is supported by all reviewers and author rebuttals clarified most of review concerns. Thus an accept is recommended.

**Justification For Why Not Higher Score:**

Some of the reviewers are only mildly supportive.

**Justification For Why Not Lower Score:**

There are consistent review supports.

---

### Decision · Program_Chairs · 2024-01-16

Accept (poster)